# The CPGs for Limbed Locomotion–Facts and Fiction

**DOI:** 10.3390/ijms22115882

**Published:** 2021-05-30

**Authors:** Sten Grillner, Alexander Kozlov

**Affiliations:** Department of Neuroscience, Karolinska Institutet, SE-17177 Stockholm, Sweden; sten.grillner@ki.se

**Keywords:** locomotion, premotor interneurons, unit burst generator network, CPG simulations

## Abstract

The neuronal networks that generate locomotion are well understood in swimming animals such as the lamprey, zebrafish and tadpole. The networks controlling locomotion in tetrapods remain, however, still enigmatic with an intricate motor pattern required for the control of the entire limb during the support, lift off, and flexion phase, and most demandingly when the limb makes contact with ground again. It is clear that the inhibition that occurs between bursts in each step cycle is produced by V2b and V1 interneurons, and that a deletion of these interneurons leads to synchronous flexor–extensor bursting. The ability to generate rhythmic bursting is distributed over all segments comprising part of the central pattern generator network (CPG). It is unclear how the rhythmic bursting is generated; however, Shox2, V2a and HB9 interneurons do contribute. To deduce a possible organization of the locomotor CPG, simulations have been elaborated. The motor pattern has been simulated in considerable detail with a network composed of unit burst generators; one for each group of close synergistic muscle groups at each joint. This unit burst generator model can reproduce the complex burst pattern with a constant flexion phase and a shortened extensor phase as the speed increases. Moreover, the unit burst generator model is versatile and can generate both forward and backward locomotion.

## 1. Introduction 

The graceful locomotor movements that we observe as a cheetah runs after a prey or a ballet dancer performs utilize most parts of the nervous system, from the spinal cord to the cortex and cerebellum. The very core of the locomotor system is, however, composed of spinal circuits under the control of locomotor command centers in the brainstem [1]. The general pattern of motor coordination during walking is ancient and similar in rodents, cats and humans and also in birds, which suggests that it had evolved already in reptiles and possibly earlier [2]. In the spinal cord, there is a locomotor network of neurons coordinating the pattern of muscle activation in each step cycle, a central pattern generator network (CPG). This network is also subject to a profound sensory control from receptors/afferents activated by the limb movements [3,4,5,6]. During a step cycle, the limb goes through four different phases, (1) the support phase, (2) the lift off phase, (3) the forward flexion during the swing phase and (4) finally, a touch down phase, which is the most demanding part, when the limb is extended to make contact with the ground with appropriate speed in relation to ground. The pattern of muscle activation is more complex than a mere alternation between flexors and extensors, and ensures that the lift off and touch down phases are adequately controlled. This complex pattern remains after a complete denervation of the afferents from the limb, and it is thus coordinated by the CPG itself [7,8]. Under normal conditions all phases are subject to modulation due to sensory input arising from the limb as it is subject to a variety of perturbations as, for example, we walk on a slope or experience obstacles. The sensory elements that influence the CPG are of critical importance for normal locomotion and have been dealt with extensively elsewhere [1,6,9]. Suffice it to mention that the extensor Golgi tendon organs activated during the support phase and receptors signaling the hip position can affect the transition from stance to swing and thus has an impact on the CPG. This review is, however, dedicated specifically to the limb CPG and particularly that of mammals, and we will not consider the important sensory aspects below. 

The forms of locomotion can be varied in many ways in humans, as well as other mammals. For instance, one can walk forwards, backwards, sideways, tiptoeing or crouching. This means that there is a need for changing, for example, the coordination between the hip and the lower leg when changing from the forward to the backward direction. It is reasonable to assume that there is not a separate locomotor CPG for each of these possibilities, but rather that the CPG needs to be versatile, and be able to coordinate each of these motor patterns. 

To understand the intrinsic operation of a CPG will require detailed information of not only which neurons are part of the CPG, but also their membrane properties and the synaptic interaction between subtypes of interneurons and finally how the CPG is turned on from the brainstem. Easy to state, but more difficult to deliver, given that this is a technically very demanding task. It has been largely solved regarding the CPGs of the zebrafish, lamprey and frog tadpole. It has, however, been difficult to reach a similar degree of understanding in the CPGs controlling limbs since the network needs to be more complex, controlling different groups of muscles in a precise manner. In Section 3 we will discuss what is known about different aspects of the limb CPG of mammals mostly from experiments on cat, rat and mouse. In Section 2, we will consider the design of the swimming CPGs, which is largely understood, and subsequently discuss the limb CPGs, what is known–the facts–and then an evaluation of conceptual schemes analyzed through detailed modelling [10,11,12,13,14].

## 2. The CPG in Lamprey, Zebrafish and Tadpole

In the zebrafish, lamprey and frog tadpole the intrinsic function of the CPG providing alternation in a given segment is in principle understood [1]. The recent data from the El Manira laboratory has shown that there are three subtypes of glutamatergic excitatory interneurons (V2a-type), projecting directly to slow, intermediate and fast motoneurons [15,16,17,18]. The “slow” interneurons projecting to the slow motor neurons have the lowest threshold for activation from descending drive signals and a tendency to oscillate when depolarized, due to their intrinsic membrane properties (Figure 1 and Figure 2D). Within the slow interneuron population there is mutual excitation that reinforces and coordinates the activity in this subpopulation of interneurons, which provides the rhythmic excitatory drive to the slow motoneurons. A further increase in the descending drive signal will recruit the “intermediate” interneuron population. In addition, these interneurons receive excitation from the slow interneuron population, which allows for synchrony. The same applies with a further increased descending drive, when the “fast” interneuron population can be brought in to generate swimming at the highest speeds of locomotion. The burst termination is due to intrinsic membrane properties. Inhibitory interneurons with a commissural action are also brought in that inhibit premotor interneurons and motor neurons on the contralateral side and ascertain that when the CPG on one side is active the other side will be inhibited and vice versa. Previous less-detailed work on the lamprey and frog tadpole had shown a similar organization with populations of excitatory glutamatergic premotor interneurons responsible for providing the excitation of motoneurons and reciprocal inhibition [19,20]. The segmental lamprey network, as shown in Figure 2B, depends on pools of interacting excitatory interneurons (E in Figure 2B), which are activated by a descending drive, but also through mutual excitation between the E-interneurons (Figure 2E). The network has been simulated in great detail (Figure 2A) with the different neurons having properties similar to their biological counterparts and with biological variability introduced. The cellular mechanisms that determine burst termination are shown in Figure 2D. Essentially, during the burst the level of Ca^2+^ will increase intracellularly, which will activate calcium-dependent potassium channels, which will gradually hyperpolarize the E-interneurons. A segmental population of 100 E-interneurons generates burst activity (Figure 2F) when submitted to descending excitatory drive [12,13,21]. The alternation between the left and right side is achieved by inhibitor interneurons. The intersegmental coordination based on available connectivity has also been simulated as detailed in the references provided above.

During swimming a rostrocaudal wave is transmitted along the length of the body that pushes the animal forward through the water, and in the spinal cord there is a corresponding rostrocaudal wave of activity that results from an interaction between the segmental networks. As shown in the lamprey, a somewhat higher excitability in the lead-segment will result in a constant phase lag along the spinal cord (Figure 2F). This mechanism will produce forward swimming, but also backward swimming if the lead segment is located in a caudal position [12,13,22].

## 3. The Hindlimb CPG in the Lumbosacral Spinal Cord

The mammalian CPG is more complex in that it needs to coordinate all the different muscles in the limb. Furthermore, the different CPGs of the four limbs can be coordinated in many different ways, as in walking, trotting and galloping. Let us now consider the CPG of a single limb. The hindlimb CPG has by far received the most attention experimentally over several decades, primarily in the cat and then in rodents (mice). The early work on the cat showed that the spinal cord could generate the complex motor pattern [7,8] even without any afferent input. The CPG could be released by activation of noradrenergic precursors [23,24,25]. To attempt to understand how the CPG is designed, recordings of rhythmically active interneurons were performed, and it was shown that inhibitory premotor neurons mediating the reciprocal inhibition between flexor and extensors, the so-called 1a interneurons, were rhythmically active and thus part of the CPG network [26,27,28,29]. Further, the Renshaw cells were active and receiving input from the CPG. However, the likelihood that one should be able to obtain sufficient rigorous knowledge of interneuronal connectivity was very low, particularly with the techniques available in the seventies. This was the reason why one of us (SG) shifted over from cat to lamprey many years ago, which allowed for paired recordings between pre- and postsynaptic neurons in the CPG. 

The situation changed radically some decades ago, when it became possible to study the CPG in action in the isolated mouse spinal cord [30,31], combined with the fact that molecular techniques entered the scene. The definition of developmentally defined subtypes of interneurons V0 to V3, which included physiologically defined subtypes of inhibitory and excitatory interneurons and dorsal subtypes of precursor neurons, opened up a wealth of new possibilities [32,33,34,35,36,37]. Subtypes of interneurons could be deleted and/or optogenetically activated or inhibited, while the behavioral effects on the locomotor network could be investigated. 

Despite these new possibilities and a dedicated effort of many prominent research groups such as those of Martyn Goulding, Ole Kiehn, Sam Pfaff, Tom Jessel, Rob Brownstone and many more, it has not yet been possible to crack the code of the limb CPG. The periodic inhibition of flexors is mediated by V1 interneurons and of extensors via V2b interneurons. A simultaneous inactivation of both groups abolishes the flexor–extensor alternation and makes them discharge in synchrony, but the rhythmic bursting still continues. This indicates that the burst generation is not crucially dependent on inhibition, which experiments with blockade of glycinergic inhibition had also suggested [38]. In the frog tadpole, recent experiments have indicated the burst generation requires commissural inhibition [39], however, in contrast to the situation in lamprey [40].

A wealth of information has been gathered concerning different subtypes of excitatory interneurons that when interfered with may either lower or make the rhythm more irregular. For instance, V2a interneurons provide excitation of motoneurons but is thought not to initiate the rhythmic burst activity as they do in the zebrafish. A population of non-V2a interneurons that expresses Shox2 has recently come in focus, which is composed by electrically coupled excitatory interneurons [41,42] and they may contribute to the generation of rhythmic bursting and provides excitation of V2a interneurons. An excellent and more detailed account for the activity in different subtypes of interneurons is provided by Rancic and Gosgnach [43] that are part of this volume.

The overall conclusion so far is that we have information on many subtypes of interneurons, but no overall understanding of the integrated CPG network that coordinates the whole limb with its complex motor pattern, except that V1 and V2b interneurons are required for the generation of the flexor–extensor alternation. More knowledge is required regarding connectivity between interneurons, synaptic properties and membrane properties. There is also little knowledge of the interaction between the premotor interneurons located in different segments along the spinal cord that are engaged in the control of different muscle groups.

## 4. Is the Ability to Generate Rhythmic Activity Distributed within the Lumbosacral Spinal Cord?

One important question to consider is if the rhythm-generation in the isolated spinal cord of the mouse locomotor activity is located in one or two segments, or if it is distributed along the entire lumbosacral spinal cord. Locomotor activity can be induced pharmacologically, by stimulation of descending pathways or optogenetically (channelrhodopsin (ChR2)) by activation of glutamatergic interneurons, including not only V2a interneurons. Hägglund et al. [44] could show that shining light on the entire spinal cord elicited coordinated locomotor activity, while light limited to only one side of the spinal cord produced coordinated locomotor bursting only on the side exposed to light. Furthermore, when the light was limited to only one segment burst activity was produced in only that segment. Furthermore, by recording activity in single flexor and extensor nerves, they could show that exclusive flexor or extensor activity could be induced separately. Thus, discrete bursting could be induced in either a flexor or an extensor. The ability to generate locomotor-related burst activity is thus distributed along the spinal cord and not limited to one part of the spinal cord. Burst activity can be generated independently in either flexors or extensors. 

These findings agree with earlier findings reported in the cat [10] showing that the spinal cord could be divided into two parts, both of which could be made to generate burst activity, and that flexor bursting could occur with tonic activity in extensors or vice versa. This led Grillner [10,45] to suggest a conceptual scheme, the unit burst generator model (UBG-model), in which each group of synergists at the different joints would be by a “unit burst generator”. Each unit CPG would be comprised by the interneurons that are presynaptic to one group of close of synergists such as the hip extensors or the ankle flexor. The different unit CPGs would interact by reciprocal inhibition as between the hip flexor UBG and that of the hip extensors, or by mutual excitation between the UBGs of hip and knee extensors. Moreover, the more complex patterns of muscle activity that occur in knee-flexors and toe extensors at the transition between main flexors and extensors at the hip could also be accounted for. We will discuss simulations of the UBG further below.

## 5. Facts regarding the Limb CPG for Locomotion—Summary 

The focus of this brief account is the mammalian limb CPG; although not understood in all aspects, we feel that it is important to briefly summarize facts regarding the different parts of the CPG jigsaw puzzle—which parts are in place, and which are undefined. Below, we will consider all interneurons being presynaptic to the motoneurons or upstream that contribute to the generation of the motor pattern as part of the locomotor CPG. Within the family of interneurons some may be more important for generating the frequency of bursting and others for inhibiting or activating a specific muscle group. We will not subdivide the CPG in one layer concerned with rhythm generation and a second downstream layer providing the motor pattern, because they operate, as far as I can see, as an integrated whole.

The CPG produces a motor pattern that includes flexor and extensor alternation and it activates in addition motoneurons with a burst in the transition between extensors and flexors as the knee-flexors (lift up) and the conversely at touch down (the E1 phase).The ability to induce locomotor bursting is distributed along the lumbosacral spinal cord on each side.Bursts can be induced independently in flexor and extensor groups in one part of the spinal cord.Motoneurons receive glutamatergic excitation alternating with glycinergic inhibition in the inactive periods.V2a interneurons contribute to motoneuron excitation and they represent a family of interneurons with many subpopulations each of which target a separate group of motoneurons.Shox2 (non-V2a) interneurons are electrically coupled and may contribute to bursting and to activation of V2a interneurons. Shox2 interneurons are rhythmically active in phase with extensors or flexors.HB9 (homeobox) interneurons are rhythmically active and provide monosynaptic excitation to motoneuron, but do not appear to contribute to the rhythm generationMonosynaptic glycinergic inhibition of motoneurons is mediated to flexors via V1 interneurons and to extensors via V2b interneurons, some of which are also referred to as 1a interneurons. The flexor and extensor groups of 1a interneurons are further subdivided in subgroups each acting on a limited set of synergist motoneurons (Eccles and Lundberg 1958).V1 and V2b groups include in addition to 1a interneurons a variety of inhibitory neurons including Renshaw cells.Deletion of V1 and V2b interneurons abolishes alternation between flexors and extensors while synchronous burst activity is maintained.

## 6. Conceptual Schemes—Models—Fiction Based on Facts

### 6.1. The Unit Burst Generator Model for the Locomotor CPG

The network for a single limb needs to be flexible, and to allow the generation of locomotor movements with different limb configurations such as walking forwards and backwards and more subtle modifications. 

A conceptual model of the limb CPG that is compatible with the available facts is the UBG model [10,45,46]. Figure 3 (left) illustrates that locomotion results from the interaction between the four limb CPGs, and further suggests that the limb CPG can be subdivided into a set of UBGs, each controlling a group of close synergists such as hip flexors or knee extensors. Each UBG is composed of a pool of excitatory neurons that can isolation generate burst activity if receiving an excitatory input. With current knowledge (see above) such a UBG could be composed of V2a and Shox2 (non-V2a; see above) interneurons that could independently generate the burst pattern. 

In the lamprey, segmental network presumed V2a interneurons can generate the burst pattern without inhibitory interaction. They would be analogous to a UBG. To explore this possibility, we have used detailed simulations that have been performed on the lamprey locomotor network [12,21], which were based on simulations of 100 excitatory interneurons in each segment. The simulations provided detailed representations of the somato-dendritic properties including subtypes of ion channels (Figure 2A), synaptic interaction including both AMPA and voltage-dependent NMDA channels (Figure 2C). The burst termination would depend on cellular mechanisms, including accumulation of Ca^2+^ and Na^+^ during the burst that activate Ca ^2+^ and Na^+^-activated potassium channels (Figure 2D). The unit CPG would be composed of pools of interacting excitatory interneurons (Figure 2E) and within each pool of interneurons the variability within the population was introduced which is an important factor which provides increased range of stable bursting. The intersegmental coordination is also illustrated for forward and backward swimming (Figure 2F).

To simulate the limb CPG, we have utilized the lamprey segmental core of E-interneurons that provide independent bursting as UBGs [21]. We assume reciprocal inhibition between flexor and extensor UBGs at each joint (see scheme in Figure 3) similar to the crossed inhibition in the lamprey network. In addition, we have considered the knee flexors (KF) that are active in the transition between support and swing phase (lift off) and let them receive inhibition from the flexors acting at the hip and ankle. This will relieve the KF-UBG from inhibition in the transition phase and discharge a burst at liftoff. The same is true for the toe extensors that are activated specifically when the limb makes contact with the ground. The EDB-UBG is indicated to receive inhibition from both flexor and extensors and thereby be active in the transition. Figure 3 shows the coordinated motor pattern produced by the network when simulated with all UBGs activated. Below is indicated the activity of the interneurons in the different unit CPGs. Note the alternation between flexors and extensors, and also knee flexors and toe extensors are active in the transition phase. This scheme captures most aspects of the mammalian locomotor pattern. 

During locomotion the flexion phase remains constant while the extension phase shortens progressively as the speed of locomotion increases. The activity of the UBG-network depends on the excitatory drive corresponding to reticulospinal neurons and indirectly from the mesencephalic locomotor center (MLR, Figure 3). If the drive to the flexor UBGs was kept constant, while that to the extensor UBGs was progressively increased, there was a progressive shortening of the extensor burst duration, as the flexion phase remained constant (Figure 4A). In this sense the UBG network reproduces the flexor and extensor asymmetry observed during locomotion in rodents, cats and humans [47,48].

When walking backwards, the coordination between the hip and the lower leg must be reversed, the liftoff face will then be in a rostral limb position and the touch down in a caudal position opposite to forward locomotion. The flexibility of the UBG network can account for this. If we hypothesize that one can switch the connectivity between the hip extensor UBG to that of the knee UBG (Figure 3, middle), from excitatory interaction to inhibition the phase relation would change as shown in Figure 4C. Note that now the hip flexors are active together with knee extensors. There are several examples of networks with a dynamic shift of connectivity between excitation and inhibition within the spinal cord [49,50]. 

A UBG-model is most likely a simplification, but it does show that kernels of unit CPGs in considerable detail can reproduce the complex motor pattern of the mammalian limb during locomotion. It can reproduce the asymmetry between flexors and extensors as the speed increases, and it can easily produce a flexible motor coordination as seen in forwards and backwards locomotion. Thus, it is possible to consider a fairly simple and flexible design of the limb CPG that is compatible with the facts so far available. In addition, it is versatile, a characteristic of the mammalian limb CPG. To what a degree it will apply will need more research of a demanding type to be resolved. A similar organization has been suggested for the stick insect with independent circuits operating at each joint [51].

### 6.2. Two Level Models

Several early reports had indicated that lesions of the upper lumbar segments innervating hip flexors resulted in an abolition of locomotor activity, which lead to the supposition that there was only one rhythm generator, despite the fact that it had been shown that also caudal segments can generate rhythmic activity [10] and the more recent demonstration in the rodent spinal cord referred to above [44]. Many researchers still favor a model with only one rhythm generator. Rybak et al. [14] have modelled such a network with a reciprocally coupled extensor and a flexor burst generator [14,52,53]. The central rhythm generator (RG) sets the coordinated burst frequency and only at that level a change in pace can be achieved. The RG in turn is thought to activate a pattern generating network (PG) which consists of a set of excitatory and inhibitory premotor interneurons. So far, this network has been modelled to generate rhythmic alternating activity, but not the finer details of the motor pattern. A question to consider is also the fact that the locomotor network is affected by sensory signals from the moving limb, and this form of control is also included in their simulations. 

One further argument that has been used to support this single RG model is that during rhythmic continuous bursting, a deletion of one burst in one muscle group can occur while the other muscles continue to be active without any change. This could easily be explained if one had only one RG that was dictating the rhythm, but it could equally well be explained with the UBG model. In fact, in the UBG configuration, one could think that the hip UBGs could have a role of having a higher level of drive and thereby represent the lead segment and the UB at other joints would be entrained by the hip UBGs under some conditions. The RG/PG concept consists of one rhythm generator with the activation/inhibition of the different muscle groups handled downstream by PG circuits. In contrast, the UBG model has rhythm and pattern generation integrated and the hip UBGs are responsible for activation of flexors and extensors at the hip, and will also affect the lower limb UBGs.

Another aspect to consider is whether the CPG network is used exclusively for locomotion or if parts of the network can be used in the context of other patterns of behavior. The first example to consider is an integrated pattern such as that used during the scratch reflex when the hindlimb is used to target a specific area on the skin which is itching. An additional possibility is whether small parts of CPG network can be selectively activated from the forebrain to elicit volitional movements of parts of the limb, for instance, moving only the ankle or wiggling the big toe [45]. It would seem useful to utilize the preformed circuitry rather than to recreate new circuits, but this needs to be explored.

### 6.3. The Swimming and Walking Motor Patterns Can Be Accounted for by One CPG Network

The swimming motor pattern is, in contrast to over ground locomotion, characterized by very short extensor burst and longer flexor burst in mice (3). This is most likely explained by the fact that the mechanical resistance to the forward movement of the limb in the flexion phase will necessarily be much greater in water than in air. When, however, the muscle spindle afferents have been eliminated the motor pattern reverts to that of walking. This case shows that the mouse uses the same “central” motor program for both walking and swimming, but in the latter case the extreme mechanical difference of the movement resistance allows afferents, perhaps from the hip, to prolong the flexion phase by action on the CPG [54].

## 7. Conclusions

Whereas the swimming CPGs are largely understood, the limb CPG remains enigmatic. It is clear that V2b and V1 interneurons are required for generating the alternation between flexors and extensors and after a deletion of these interneuron populations, flexors and extensors burst together. The generation of bursting activity may depend on Shox2 interneurons that activate V2a interneurons and in turn motoneurons and other groups such as HB9 may contribute. The ability to generate burst activity is distributed along the lumbosacral spinal cord and local circuits can activate flexors or extensors in one segment independently. It is shown that the motor pattern underlying locomotion could be explained by the unit burst generator model. In addition, it can account for the asymmetry between flexors and extensors as the speed increases and it is a versatile conceptual model that allows for, for example, forward and backward locomotion. Although a model is compatible with available experimental data, it may of course reflect only part of the spinal cord reality.

## Figures and Tables

**Figure 1 ijms-22-05882-f001:**
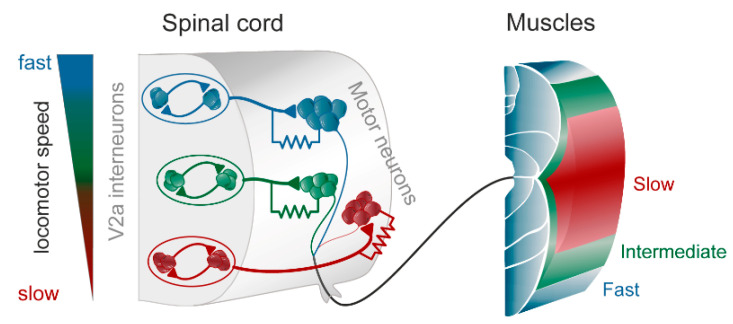
The modular zebrafish central pattern generator (CPG). As described in the text, the excitatory V2a interneurons are further subdivided into three modules, each controlling fast (blue), intermediate (green), and slow (red) motoneurons and muscle fibers. Each module consists of excitatory interneurons, some of which also have gap junctions in their synapse on motoneurons (as indicated by the resistor in the drawing). The gap junctions play an important role in providing feedback between motoneurons and interneurons, thereby amplifying the chemical synaptic transmission when depolarized. To the left is indicated that at slow speeds the slow (red) module is recruited, and with increasing speed the intermediate (green) and the fast (blue), respectively. Modified from Grillner and El Manira (2020) [1].

**Figure 2 ijms-22-05882-f002:**
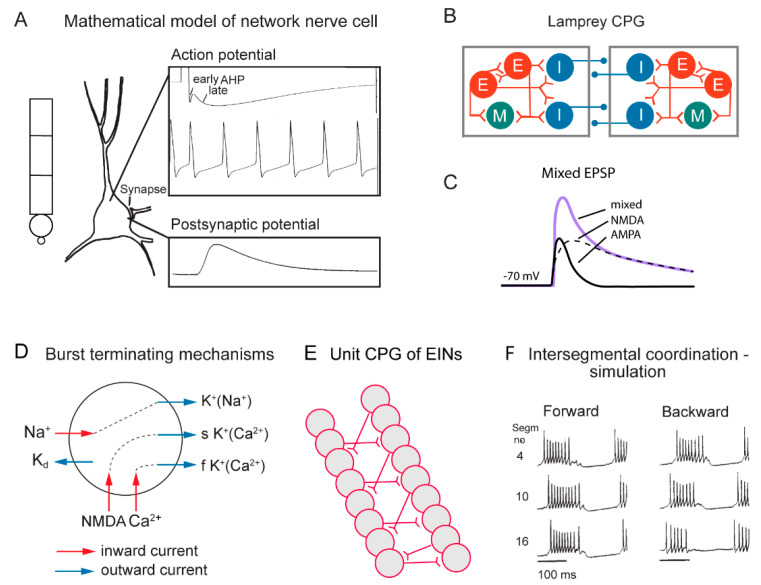
Cellular properties explored in simulations of the lamprey spinal locomotor networks. (**A**) The morphology of the CPG neurons is captured using a five-compartment model consisting of an initial segment, a soma and a dendritic tree. Active ion currents are modeled using a Hodgkin–Huxley formalism. Both ion channels involved in spiking behavior (Na^+^, K^+^) and slower Ca^2+^ or potassium-dependent processes (K_Ca_, K_Na_) are modeled based on available data. Spiking behavior of a CPG neuron. Spike frequency can be regulated by the AHP (afterhyperpolarization). (**B**) The segmental locomotor network in lamprey. Red, excitatory interacting interneurons excite motoneurons (green) and commissural inhibitory interneurons (blue) that inhibit all neurons on the contralateral side. (**C**) Fast synaptic transmission is included in the form of excitatory glutamatergic (AMPA and voltage-dependent NMDA) and inhibitory glycinergic inputs. (**D**) Main ionic membrane and synaptic currents considered to be important during activation within the CPG network. Slower processes can cause spike frequency adaptation, such as Ca^2+^ accumulation during ongoing spiking and resulting activation of K_Ca_. (**E**) Unilateral CPG activity can be evoked in local networks of excitatory interneurons (EINs). This basic EIN network can sustain the rhythm seen in ventral roots in vitro during evoked locomotor activity. The left–right alternating activity requires the presence of contralateral inhibition provided by glycinergic interneurons (see below). (**F**) The burst activity along the simulated spinal cord is illustrated in segments 4, 10 and 16. Note the sequential activation of the different segments.

**Figure 3 ijms-22-05882-f003:**
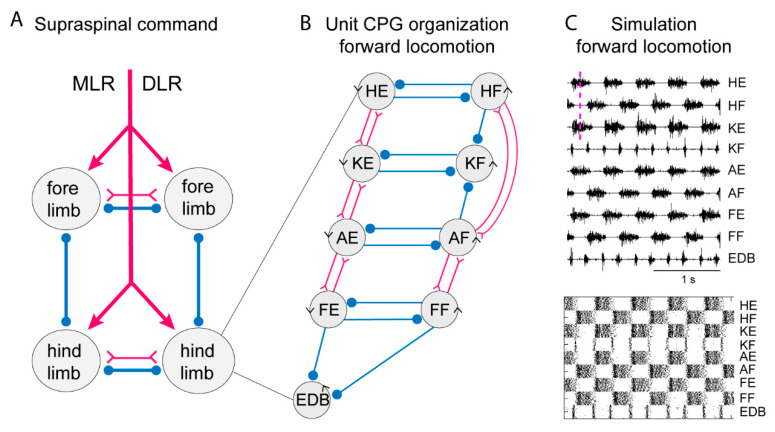
Systems of interacting unit CPGs – intralimb coordination – forward locomotion. (**A**) The diagram to the left shows the four limb CPGs of a tetrapod and possible modes of coordination (in phase or alternation for fore- and hindlimbs, respectively). The CPGs are turned on by the descending drive from the MLR (mesencephalic locomotor region) or the DLR (diencephalic locomotor region). (**B**) Within each limb CPG there is most likely a further subdivision in unit CPGs controlling the synergists at one joint, such as hip (H), knee (K), ankle (A) and foot (F) extensors (E) or flexors (F). EDB (extensor digitorum brevis) has a particular pattern. The normal pattern of activity results from the interaction between the different unit CPGs at different joints. The advantage is that the unit CPGs may be recombined as in backward or forward walking, in the same way as with the different limb CPGs that can be recombined in the different gaits. Circles indicate inhibition, and forks/triangles excitation. (**C**) Exploratory simulation of locomotor activity, with a network arranged as in the diagram in which each unit CPG is designed in a similar way to the lamprey unit CPGs consisting of 100 excitatory interacting neurons–and the interaction between the 9-unit CPGs arranged as in the middle diagram. The output of the model network captures essential features of the locomotor output.

**Figure 4 ijms-22-05882-f004:**
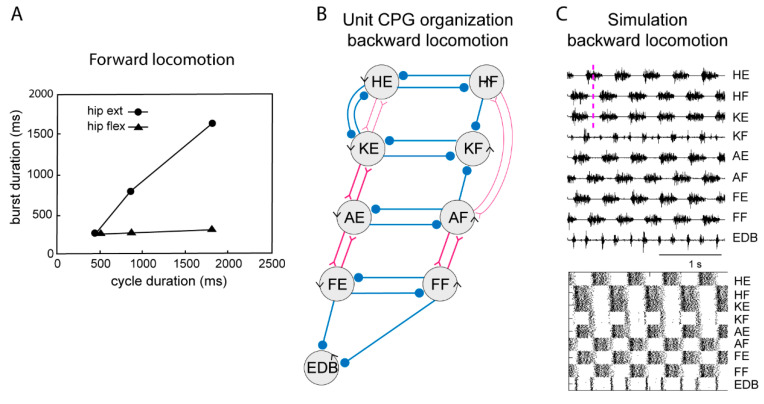
Systems of interacting unit CPGs – intralimb coordination – backward locomotion. (**A**) The graph shows that the flexion phase can be kept constant at different cycle durations, while the extension phase can vary markedly, when the excitatory drive to the extensor unit CPGs is varied, although the drive to the flexor unit CPGs is kept constant. (**B**) Note that the connectivity pattern between the hip unit CPGs and the lower limb unit CPGs has been modified. Weakened synaptic projections are shown with thin lines. (**C**) Same representation of the motor pattern to the different muscles as in Figure 3, during simulated backward locomotion. Below is indicated the activity in individual interneurons.

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
