# Peer review of "The CPGs for Limbed Locomotion–Facts and Fiction"

_ijms, 2021, doi:10.3390/ijms22115882_

Round 1

Reviewer 1 Report

This is an interesting review that obviously is written to fit into a special issue and thus will relate to other articles in ways that can’t be appreciated by examining only this work. Overall, the key concepts put forward are solid and of interest, and nicely demonstrate that the UBG model (or using UBGs to model CPGs) can account for many of the key features of stepping.  It nicely relates the lamprey system to the more complex limbed mammalian organization. There are several minor, and a few fairly significant concerns that, if addressed, will hopefully improve the review and make it more accessible and informative to the reader who is not already a CPG afficionado.

Major:

In general, sections 1 and 2 are too short and are lacking information that is needed to appreciate and understand the figures, which are incredibly important to the overall goals of the review. This is partly due to the use of acronyms (see below), but also this section assumes too much knowledge for the reader.

Numerous acronyms/initials need to be spelled out when first used. EINs, AHP, MLR, DLR.

The crux of the story (lines 274-303) relies heavily on parts of Figures 3 & 4. It may be helpful to take the key parts of these figures, and make them larger, as another figure to illustrate the key points (including perhaps some vertical lines to illustrate the key features of the step cycle). This would also help the for the discussion of backward walking with parts of Figure 4. You are arguing that the UBG model reproduces key features of actual locomotor output, but it is not easy to recognize this as the article and figures now stand.

Minor:

The discussion of “a single RG” vs a distributed network of UBGs (or CPGs) is, perhaps, incomplete from at least one perspective. If you put forward the possibility of “lead UBGs” (i.e. for the hip), with higher level drive, you have essentially argued for a two-layer system with RG and PG circuitry. Similarly, a number of authors have suggested that rhythmic elements are present along the entire neuraxis, but when intact, these are led by (and are subservient to) key segments (L1/2 for the hindlimbs and C5/6 for the forelimbs), which play the role of RG circuitry. Perhaps this concept should be included.

Finally, swimming and stepping in rodents have some characteristics that suggest they have distinct control mechanisms and that the circuitry involved is not entirely overlapping. Since swimming in the lamprey is your model system, perhaps also including the concept of limbed swimming would be appropriate. Swimming involves a short extensor burst and a longer flexion burst, quite different from stepping. It is also a bipedal activity in rodents, which carries interesting control characteristics.

Some suggested changes:

Line 30. “in place already” is a very loose way of referring to this.

Line 39. “ascertains” could probably be improved by saying “ensures”

Line 43. Walk “uphill” and “downhill”, or on a slope.

Line 70. Why would the involvement of gap junctions be called “a new role”? Do you mean new to the zebrafish, new to us (scientists)?

Line 128. “a further increase in descending drive”.

Line 200. The description of the Hagglund work is lacking some important details. It is implied, due to the preceding sentence, that this is optogenetic stimulation of V2a neurons, but that is not completely clear.

Lines 213-218. These are key components of this section and unfortunately, are not very clear.

Line 222. I think it would suffice to say “upstream of motoneurons” or “presynaptic to motoneurons”, but not both.

Line 226. This conclusion seems to be in conflict with McCrea and Rybak who concluded that rhythm and pattern can be independently modulated (but of course are interconnected). This may not be the place to tackle this subtlety, but acknowledging the potential for independent modulation of rhythm and pattern is probably important. This feeds into the “single RG” comment above.

Line 229. This is a confusing contribution to the list of facts.

Line 256. It is assumed you mean “…that can, in isolation, generate burst activity…”

Author Response

Comments and Suggestions for Authors

This is an interesting review that obviously is written to fit into a special issue and thus will relate to other articles in ways that can’t be appreciated by examining only this work. Overall, the key concepts put forward are solid and of interest, and nicely demonstrate that the UBG model (or using UBGs to model CPGs) can account for many of the key features of stepping.  It nicely relates the lamprey system to the more complex limbed mammalian organization. There are several minor, and a few fairly significant concerns that, if addressed, will hopefully improve the review and make it more accessible and informative to the reader who is not already a CPG afficionado.

We are grateful for the appreciation!

Major:

In general, sections 1 and 2 are too short and are lacking information that is needed to appreciate and understand the figures, which are incredibly important to the overall goals of the review. This is partly due to the use of acronyms (see below), but also this section assumes too much knowledge for the reader.

We agree and have added many lines of new text in section 1 and 2

Numerous acronyms/initials need to be spelled out when first used. EINs, AHP, MLR, DLR.

Done

The crux of the story (lines 274-303) relies heavily on parts of Figures 3 & 4. It may be helpful to take the key parts of these figures, and make them larger, as another figure to illustrate the key points (including perhaps some vertical lines to illustrate the key features of the step cycle). This would also help the for the discussion of backward walking with parts of Figure 4. You are arguing that the UBG model reproduces key features of actual locomotor output, but it is not easy to recognize this as the article and figures now stand.

We agree and have introduced a red vertical line In Figure 3 and 4 to indicate the difference between the coordination of hip and knee extensors, and use the full width of the page for the two figures, and believe that this will be sufficient to illustrate the key-points.

Minor:

The discussion of “a single RG” vs a distributed network of UBGs (or CPGs) is, perhaps, incomplete from at least one perspective. If you put forward the possibility of “lead UBGs” (i.e. for the hip), with higher level drive, you have essentially argued for a two-layer system with RG and PG circuitry. Similarly, a number of authors have suggested that rhythmic elements are present along the entire neuraxis, but when intact, these are led by (and are subservient to) key segments (L1/2 for the hindlimbs and C5/6 for the forelimbs), which play the role of RG circuitry. Perhaps this concept should be included.

I have added the following text

The RG/PG concept consists of one rhythm generator with the activation/inhibition of the different muscle groups handled downstream by PG circuits. In contrast the UBG model has rhythm and pattern generation integrated and the hip UBGs are responsible for activation of flexors and extensors at the hip, and also affect the lower limb UBGs.

Finally, swimming and stepping in rodents have some characteristics that suggest they have distinct control mechanisms and that the circuitry involved is not entirely overlapping. Since swimming in the lamprey is your model system, perhaps also including the concept of limbed swimming would be appropriate. Swimming involves a short extensor burst and a longer flexion burst, quite different from stepping. It is also a bipedal activity in rodents, which carries interesting control characteristics.

We have added a section 6.3 on the interesting difference between swimming and walking!!

Some suggested changes:

Line 30. “in place already” is a very loose way of referring to this.

done

Line 39. “ascertains” could probably be improved by saying “ensures”

done

Line 43. Walk “uphill” and “downhill”, or on a slope.

done

Line 70. Why would the involvement of gap junctions be called “a new role”? Do you mean new to the zebrafish, new to us (scientists)?

Replaced by important

Line 128. “a further increase in descending drive”.

done

Line 200. The description of the Hagglund work is lacking some important details. It is implied, due to the preceding sentence, that this is optogenetic stimulation of V2a neurons, but that is not completely clear.

done

Lines 213-218. These are key components of this section and unfortunately, are not very clear.

Replaced by: The ability to generate locomotor related burst activity is thus distributed along the spinal cord and not limited to one part of the spinal cord and burst activity can be generated independently in either flexors or extensors.

Line 222. I think it would suffice to say “upstream of motoneurons” or “presynaptic to motoneurons”, but not both.

done

Line 226. This conclusion seems to be in conflict with McCrea and Rybak who concluded that rhythm and pattern can be independently modulated (but of course are interconnected). This may not be the place to tackle this subtlety, but acknowledging the potential for independent modulation of rhythm and pattern is probably important. This feeds into the “single RG” comment above.

Line 229. This is a confusing contribution to the list of facts.

I suppose you refer to the introductory few lines with the definition of what I include as a CPG – which in my mind is important to mention

Line 256. It is assumed you mean “…that can, in isolation, generate burst activity…”

done

Submission Date

29 April 2021

Date of this review

05 May 2021 16:26:00

Reviewer 2 Report

In this manuscript, the authors summarize the current understanding of the neural networks underlying locomotion in swimming vertebrates and tetrapods, and present a model for the mammalian locomotor CPG using the unit burst generator.  The topic is important and periodical updates published by the first authors in the past have served as impactful resources for the field of CPGs.  However, I have one major issue and many minor issues with this review article that are listed below:

Major:

The authors present simulation results in this article.  However, there is not sufficient detail provided to evaluate the simulation.  If the authors are to include this simulation, they should include sufficient details of how the model was built.

Minor:  

  1. Not a complete picture of the field of the research described:
  • Line 169 to 176, The authors cite work carried out by Bracci et al., concluding that inhibitory inputs are not crucial for the rhythmic bursting based on the persistence of rhythmic activity after the inactivation of the inhibitory interneurons. However, it is important to also cite the work of Moult PR, Cottrell GA, and Li W-C (Neuron 2013) that argues that these results may be obtained due to compensatory (or homeostatic) changes in neurons after the inactivation.  

  1. Too many vague, incomplete, run-on sentences, and missing definition makes it difficult to read.

  1. Insufficient detail provided to understand what is currently known in the field
  • Line 123 – “.. mutual excitation that reinforces…”  What kind of mutual connection excites a population of interneurons?
  • Line 127 – how do the slow interneuron projections to the intermediate neurons generate synchrony at an intermediate frequency?

  1. Some shortcuts or wording used in the manuscript make the description either inaccurate or difficult to understand. Similarly, there are vague, incomplete, and run-on sentences.  The authors should go through the manuscript and enhance the clarity.  Note that the followings are some examples of these issues and not a complete list. 
  • Line 13 to 14 “It is unclear how the rhythmic bursting is generated but Shox2, V2a and HB9 interneurons will contribute.” This sentence is hard to understand because the first and the second parts are incongruous.  Do you mean that “there is some evidence showing that these interneurons may contribute to rhythm generation”? 
  • Line 16 “… each group of close synergists at each joint”. Do you mean synergistic muscle groups?
  • Line 69 “…interneurons, some of which also have gap junctions.” It should read “interneurons, some of which are connected to motor neurons via gap junctions.”
  • Line 121 – “… for activation from descending drive signals and tendency to oscillate, when activated …”.  It should read “their membrane potential tends to oscillate when depolarized”. 
  • Line 180 – “A Shox2 population of non-V2a interneurons…” It should read “a population of non-V2a interneurons that expresses Shox2.  If they are going to use this short-hand, they should spell it out the first time they use it. 
  • Line 201 “… that shining light on the entire spinal cord elicited …” Without describing that optogenetic actuators were expressed by the excitatory and inhibitory interneurons of the spinal cord, this description does not make sense.

  1. Some terminologies need to be defined
  • Line 87, “EIN” should be spelled out
  • Line 215, “unit CPG” should be defined.
  • Line 316, “rhythm generator”. How does it differ from the pattern generator?

  1. Some sentences are not well integrated into the paragraph
  • Line 325, “It has also included sensory control of the RG”. How does this piece of information relate to pattern and rhythm generation?

  1. Grammatical error, typo, or lack of consistencies
  • Line 129 “… to generate swimming at the very highest speeds…” the word “very” should be deleted.
  • Line 29 to 30 “… in rodents, cats, and humans, and also in birds, which…” should read “in rodents, cats, humans, and birds, which…”
  • Line 36 ”…and 4) finally, the most demanding …” should read “and 4) finally a touch-down phase, which is the most demanding…”
  • Line 46 ”… particulary tat of mammals. ” Should be “that”
  • Line 331 … “a role of being having a higher…” delete “being”
  • Line 152 to 155 – this long sentence should be broken into two.

  1. Section 5 is not integrated into the manuscript. Tue authors should state why it is important for the readers to be read the summary, and how these pieces of information are helpful in understanding the following sections. 

  1. There were many issues with figures
  • Figure captions are incomplete.
    • Figure captions are missing from Fig 2F
    • Figure 1 El Manira was not cited in the caption
  • Figure 2D – all the atoms (Na, K, Ca) should be ions
  • Figure 3 is cited before Figure 2,
  • Line 122 refers to the wrong figure (should refer to fig 2, not 1).
  • Three panels of figure 3 should be labeled A, B, and C.
  • Figures 3 and 4, right panel, are presumably nerve recordings on top and spike plots on the bottom, but should be clearly stated.
  • The title of figures 3 and 4 are the same. The title of the figures should be unique and more informative.
  • Figure 4A. there is no description for which one of the data points of hip extensor received increased stimulation, and the readers are left to assume the right data points are in response to the highest intensity of stimulation. 

Other:

  • Line 130 – “due to intrinsic membrane properties”. The authors should refer to the figure 2D

Author Response

Open Review

English language and style

(x) Extensive editing of English language and style required
( ) Moderate English changes required
( ) English language and style are fine/minor spell check required
( ) I don't feel qualified to judge about the English language and style

Is the work a significant contribution to the field?

Is the work well organized and comprehensively described?

Is the work scientifically sound and not misleading?

Are there appropriate and adequate references to related and previous work?

Is the English used correct and readable?

Comments and Suggestions for Authors

In this manuscript, the authors summarize the current understanding of the neural networks underlying locomotion in swimming vertebrates and tetrapods, and present a model for the mammalian locomotor CPG using the unit burst generator.  The topic is important and periodical updates published by the first authors in the past have served as impactful resources for the field of CPGs.  However, I have one major issue and many minor issues with this review article that are listed below:

We thank the reviewer for valuable comments

Major:

The authors present simulation results in this article.  However, there is not sufficient detail provided to evaluate the simulation.  If the authors are to include this simulation, they should include sufficient details of how the model was built.

We have now added further text and references.

Minor:  

  1. Not a complete picture of the field of the research described:
  • Line 169 to 176, The authors cite work carried out by Bracci et al., concluding that inhibitory inputs are not crucial for the rhythmic bursting based on the persistence of rhythmic activity after the inactivation of the inhibitory interneurons. However, it is important to also cite the work of Moult PR, Cottrell GA, and Li W-C (Neuron 2013) that argues that these results may be obtained due to compensatory (or homeostatic) changes in neurons after the inactivation.  
  • Done

  1. Too many vague, incomplete, run-on sentences, and missing definition makes it difficult to read.

We have gone over the text simplified and shortened some sentences and also added explanations for abbreviations that we have unfortunately missed. We have added further explanatory texts in section 1 and 2

  1. Insufficient detail provided to understand what is currently known in the field
  • Line 123 – “.. mutual excitation that reinforces…”  What kind of mutual connection excites a population of interneurons?
  • If a group of neurons are active and have mutual excitatory connections within the population this will reinforce the activity
  • Line 127 – how do the slow interneuron projections to the intermediate neurons generate synchrony at an intermediate frequency?
  • The burst frequency of the slow population will also increase and the synaptic interaction between the two populations will ascertain synchrony

  1. Some shortcuts or wording used in the manuscript make the description either inaccurate or difficult to understand. Similarly, there are vague, incomplete, and run-on sentences.  The authors should go through the manuscript and enhance the clarity.  Note that the followings are some examples of these issues and not a complete list. 

We have gone over the text and also explanations for abbreviations that we have unfortunately missed

  1.  
  • Line 13 to 14 “It is unclear how the rhythmic bursting is generated but Shox2, V2a and HB9 interneurons will contribute.” This sentence is hard to understand because the first and the second parts are incongruous.  Do you mean that “there is some evidence showing that these interneurons may contribute to rhythm generation”? 
  • Sentence modified
  • Line 16 “… each group of close synergists at each joint”. Do you mean synergistic muscle groups?
  • Yes – sentence modified
  • Line 69 “…interneurons, some of which also have gap junctions.” It should read “interneurons, some of which are connected to motor neurons via gap junctions.”
  • modified
  •  
  • Line 121 – “… for activation from descending drive signals and tendency to oscillate, when activated …”.  It should read “their membrane potential tends to oscillate when depolarized”. 
  • modified
  • Line 180 – “A Shox2 population of non-V2a interneurons…” It should read “a population of non-V2a interneurons that expresses Shox2.  If they are going to use this short-hand, they should spell it out the first time they use it. 
  • done
  • Line 201 “… that shining light on the entire spinal cord elicited …” Without describing that optogenetic actuators were expressed by the excitatory and inhibitory interneurons of the spinal cord, this description does not make sense.
  • Channelrhodopsin2 (chR2) added

  1. Some terminologies need to be defined
  • Line 87, “EIN” should be spelled out
  • done
  • Line 215, “unit CPG” should be defined.
  • Defined as follows Each unit CPG would be comprised by the interneurons that are presynaptic to one group of close of synergists like the hip extensors or the ankle flexors etc..
  • Line 316, “rhythm generator”. How does it differ from the pattern generator?
  • According to Rybak and others the rhythm generator sets the pace, and downstream there is a set of interneurons that distribute the excitation to all the different muscles of the limb. They refer to this set of interneurons as the pattern generating part of the network.

  1. Some sentences are not well integrated into the paragraph
  • Line 325, “It has also included sensory control of the RG”. How does this piece of information relate to pattern and rhythm generation?

 Sentence modified

  1. Grammatical error, typo, or lack of consistencies
  • Line 129 “… to generate swimming at the very highest speeds…” the word “very” should be deleted.

done

  • Line 29 to 30 “… in rodents, cats, and humans, and also in birds, which…” should read “in rodents, cats, humans, and birds, which…”

Do not agree, since birds a different from the mammalian examples

  • Line 36 ”…and 4) finally, the most demanding …” should read “and 4) finally a touch-down phase, which is the most demanding…”

done

  • Line 46 ”… particulary tat of mammals. ” Should be “that” - done
  • Line 331 … “a role of being having a higher…” delete “being” - done
  • Line 152 to 155 – this long sentence should be broken into two. done

  1. Section 5 is not integrated into the manuscript. Tue authors should state why it is important for the readers to be read the summary, and how these pieces of information are helpful in understanding the following sections. 

Introductory sentence added

  1. There were many issues with figures
  • Figure captions are incomplete.
    • Figure captions are missing from Fig 2F done
    • Figure 1 El Manira was not cited in the caption done
  • Figure 2D – all the atoms (Na, K, Ca) should be ions - They are indicated as ions
  • Figure 3 is cited before Figure 2, Not now
  • Line 122 refers to the wrong figure (should refer to fig 2, not 1). Not correct
  • Three panels of figure 3 should be labeled A, B, and C. Done
  • Figures 3 and 4, right panel, are presumably nerve recordings on top and spike plots on the bottom, but should be clearly stated.
  • The title of figures 3 and 4 are the same. The title of the figures should be unique and more informative. They should be similar except for the importance difference of forwards and backwards
  • Figure 4A. there is no description for which one of the data points of hip extensor received increased stimulation, and the readers are left to assume the right data points are in response to the highest intensity of stimulation. Explanatory text added

Other:

  • Line 130 – “due to intrinsic membrane properties”. The authors should refer to the figure 2D done

Submission Date

29 April 2021

Date of this review

19 May 2021 01:29:33
